# Bioecology of fall armyworm *Spodoptera frugiperda* (J. E. Smith), its management and potential patterns of seasonal spread in Africa

**Saliou Niassy**[1]*, **Mawufe Komi Agbodzavu**[1,2], **Emily Kimathi**[1], **Berita Mutune**[1], **El Fatih M. Abdel-Rahman**[1], **Daisy Salifu**[1], **Girma Hailu**[1], **Yeneneh T. Belayneh**[3], **Elias Felege**[4], **Henri E. Z. Tonnang**[1], **Sunday Ekesi**[1], **Sevgan Subramanian**[1]

**1** International Centre of Insect Physiology and Ecology, Nairobi, Kenya, **2** International Institute of Tropical Agriculture (IITA), Kinshasa, DR Congo, **3** DCHA/OFDA, Washington, D.C., United States of America, **4** Desert Locust Control Organization for Eastern Africa (DLCO-EA), Ethiopia

* sniassy@icipe.org

**Data Availability Statement:** Data has been stored in the stable platform Dryad and can be accessed at: Niassy, Saliou (2021), FAMEWS DATA ON FAW,

## Abstract

Fall armyworm, *Spodoptera frugiperda* (J. E. Smith) has rapidly spread in sub-Saharan Africa (SSA) and has emerged as a major pest of maize and sorghum in the continent. For effective monitoring and a better understanding of the bioecology and management of this pest, a Community-based Fall Armyworm Monitoring, Forecasting, Early Warning and Management (CBFAMFEW) initiative was implemented in six eastern African countries (Ethiopia, Kenya, Tanzania, Uganda, Rwanda and Burundi). Over 650 Community Focal Persons (CFPs) who received training through the project were involved in data collection on adult moths, crop phenology, cropping systems, FAW management practices and other variables. Data collection was performed using Fall Armyworm Monitoring and Early Warning System (FAMEWS), a mobile application developed by the Food and Agricultural Organization (FAO) of the United Nations. Data collected from the CBFAMFEW initiative in East Africa and other FAW monitoring efforts in Africa were merged and analysed to determine the factors that are related to FAW population dynamics. We used the negative binomial models to test for effect of main crops type, cropping systems and crop phenology on abundance of FAW. We also analysed the effect of rainfall and the spatial and temporal distribution of FAW populations. The study showed variability across the region in terms of the proportion of main crops, cropping systems, diversity of crops used in rotation, and control methods that impact on trap and larval counts. Intercropping and crop rotation had incident rate 2-times and 3-times higher relative to seasonal cropping, respectively. The abundance of FAW adult and larval infestation significantly varied with crop phenology, with infestation being high at the vegetative and reproductive stages of the crop, and low at maturity stage. This study provides an understanding on FAW bioecology, which could be vital in guiding the deployment of FAW-IPM tools in specific locations and at a specific crop developmental stage. The outcomes demonstrate the relevance of community-based crop pest monitoring for awareness creation among smallholder farmers in SSA.

Dryad, Dataset, https://doi.org/10.5061/dryad.
9kd51c5gg.

**Funding:** The current study was funded by USAID
through the FAO and Dr Sevgan Subramanian is
the PI at ICIPE and Dr Saliou Niassy implemented
project activities. We acknowledge support from
the European Union Delegation. We also gratefully
acknowledge the financial support for the core
research agenda of ICIPE by UK Aid from the UK
Government; the Swedish International
Development Cooperation Agency (SIDA); the
Swiss Agency for Development and Cooperation
(SDC); and the Government of Kenya.

**Competing interests:** The authors have declared
that no competing interests exist.

## 1. Introduction

Cereal crops play a vital role in the daily diets in Africa and account for up to 46% of the daily calorie consumption [1]. Maize followed by sorghum are major staple food crops grown in diverse agro-ecological zones and farming systems in sub-Saharan Africa (SSA) [2]. Cereal production in SSA is heavily constrained by different abiotic and biotic factors. Abiotic constraints are mainly related to poor soil health, poor soil fertility management, and soil nutrient depletion that arise as a result of resource degradation and erosion. Extended droughts and unpredictable rainfall patterns are climate-change-related challenges faced by growers [3, 4]. Biotic constraints encompass insect pests, such as stemborers [5, 6], striga weeds [7, 8] and diseases [9, 10]. All these constraints expose small-scale farmers to hunger and other forms of vulnerability that has been recently exacerbated with the invasion of fall armyworm (FAW), *Spodoptera frugiperda* (J. E. Smith). FAW was first reported in Africa in 2016 and adds another constraint to the complex of cereal pests in Africa [11]. The pest is native to tropical and subtropical regions of the Americas where it has long been a major agricultural problem [12]. The presence of FAW is now confirmed in 45 African countries [13, 14]. Massive yield losses, especially of maize, have been borne by farmers all over the continent. It is estimated that crops worth over USD 13 billion per annum are at risk of FAW damage throughout sub-Saharan Africa, thereby threatening the livelihoods of millions of poor farmers [15, 16].

The management of FAW in the USA is mostly based on the use of transgenic maize and chemical pesticides [17, 18]. Governments, non-govermental organizations (NGOs) and other development partners in Africa have intensified efforts to manage the invasive pest through various interventions that include pesticide application, bio-agents, pheromone traps, and push-pull cropping systems [19–21]. These interventions have been channeled to the affected farmers through extension programs, farmer field schools training, farmers associations and mass communication campaigns.

The current response in FAW management is pesticide-based and largely derived from previous experiences in the Americas, where maize cropping system is predominantly of the commercial-scale type. In contrast, maize and other cereal cropping in Africa are mainly driven by small-scale, subsistance farmers with limited resources [13]. The efficacy of a synthetic pesticide-based management strategy is not guaranteed as the pest is known to have developed resistance to a number of active ingredients, not to mention the adverse impacts of pesticides to human, non-target and beneficial organisms and the environment [22–24]. Hence, there is a need for sustainable and effective pest management technologies for FAW that could be developed through a better understanding of the pest bioecology in relation to cropping systems and practices in Africa. In North America, migration starts from Texas and Florida in the late winter or spring (February–May). Populations are noticed in the northern areas in late summer and fall, therefore the name fall armyworm [12]. In Africa, with conducive climatic conditions, FAW appears to be established [11] with a seasonal spread among regions. However, this needs to be established through field collection of temporal and spatial data on FAW population dynamics.

To understand the region-wide dynamics of FAW, data was collected through a Community-based Fall Armyworm Monitoring, Forecasting, Early Warning and Management (CBFAMFEW) initiative that was funded by the United States Agency for International Development, Office of Foreign Disaster Assistance (USAID-OFDA) and implemented through the UN/FAO in six eastern African countries (Burundi, Ethiopia, Kenya, Rwanda, Tanzania and Uganda). This initiative resulted in the collection of long-term time-series data on FAW population dynamics using pheromone traps, field-scouting and monitoring of immature stages of the pest. The FAMEWS mobile-application was used to collate field data and keep track of

other variables on cropping seasons and management practices. Pheromone traps are known to be a convenient means for monitoring and quantifying FAW adult male incidence and early warning at a given time and place [25] while field scouting and surveys with FAMEWS help in monitoring of immature stages and estimating the level of infestation, together with record-keeping systems and management practices [21]. The data that have been collected through pheromone-trap-based monitoring and field scouting protocol could be valuable in improving our understanding of FAW bioecology in Africa in relation to crop diversity and cropping systems (rotation, intercropping, and seasonality). Since FAW does not diapause, understanding its population dynamics in relation to crop phenology and its potential migration/spread in Africa is essential for effective deployment of Integrated Pest Management (IPM) techniques [26]. For example, better understanding of adult FAW movements and dynamics of its different life stages could be useful in guiding release of natural enemies for specific life stages of the pest as well as to determine other control strategies in specific locations and at a given crop stage. Hence, this study was designed to analyze long-term FAW data on FAW abundance and damage obtained through the CBFAMFEW approach promoted in East Africa for a better understanding of FAW bioecology and contribute to developing sustainable management strategies in SSA.

## 2. Materials and methods

### 2.1. Survey sites

The CBFAMFEW project was implemented in six eastern African countries (Burundi, Ethiopia, Kenya, Rwanda, Tanzania and Uganda) between August 2017 and August 2019. In each country, five maize-growing districts were selected, and in each district, 10 villages were selected for data collection, with two Community Focal Persons (CFPs) per village. The selected districts in each country for the study are presented in Fig 1, and the locations for larval scouting and trapping of FAW males in the six countries are presented in Fig 2.

### 2.2. Community-based Fall Armyworm Monitoring, Forecasting, Early Warning and Management system (CBFAMFEW)

The **CBFAMFEW** project established a network that allowed each selected village to have a trained CFP with a smartphone and a pheromone trap to collect FAW data. The CFPs were trained in the use of the FAO-developed FAMEWS mobile application (app) for data input on FAW incidence. CFPs were also trained on how to interpret data and provide timely advice to villagers and early warning of FAW attacks. Further training on regular field scouting and detecting the pest at various stages was also provided to the CFPs. To coordinate data collection and transmission, mobile phones were loaded with the FAW monitoring application app, FAMEWS. The FAMEWS app is available in Google Play in 13 languages, and was downloaded by farmers and extension workers. In total, 56 mobile phones were provided per country (1 mobile phone per village x 50 villages + 1 mobile per district officer x 5 officers + 1 mobile for the National Coordinator). Hence a total of 336 mobile phones were deployed in the six countries. Parallel to FAW population monitoring activities in East Africa, similar initiatives were conducted in Ghana, Liberia, Zambia, Mozambique and South Sudan using similar approach. Data collected from these countries using the FAMEWS app were also included with data obtained from the six eastern African countries to map the FAW density across a wider area of SSA.

### 2.3. Pheromone traps and field scouting

For monitoring adults, two universal bucket traps (Unitrap) were installed in each village. Traps baited with FAW pheromone were deployed just after planting and monitoring started

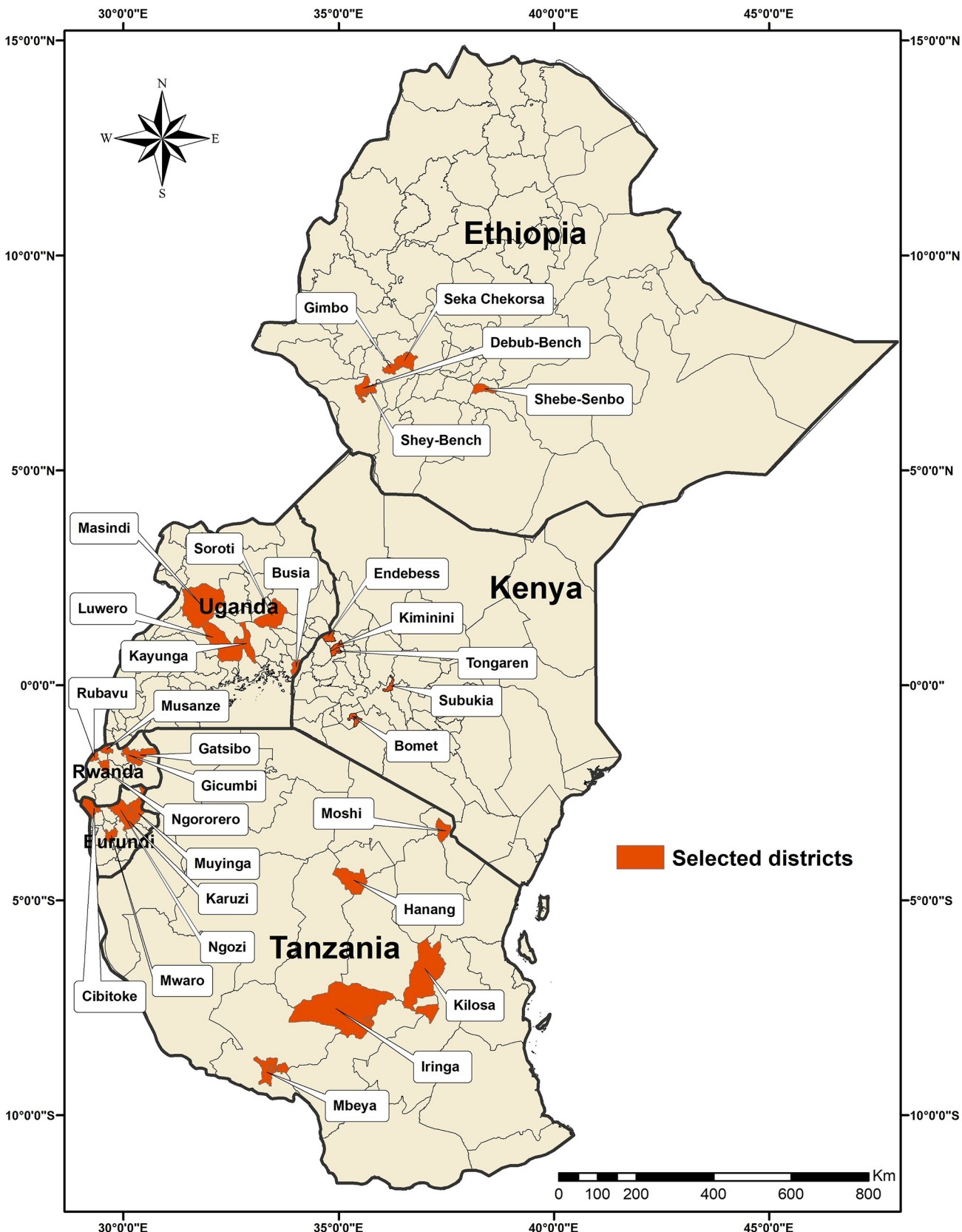

**Fig 1. Districts selected for the Community-based fall armyworm Monitoring, Forecasting, Early Warning and Management (CBFAMFEW) initiative in six eastern African countries.**

after the emergence of plant seedlings to detect the first arrival of moth pests. Traps and lure (lure blend: Z9-14Ac (81.7%); z11-16Ac (17.54%); Z7-12Ac (0.5%) and z9-12Ac (0.25%)) were supplied by Russell IPM Ltd, Unit 45, First Avenue, Deeside Industrial Park Deeside, Flintshire, CH5 2NU United Kingdom. The FAW monitoring kit also contained a toxicant which immobilized the moths once attracted to the device. One trap was placed inside the maize field and a second one outside the maize fields. The trap was hung from a suspended pole about 1.5 m above the ground, and one trap was used for every 0.5–2 ha. The pheromone lure was replaced every 3–6 weeks. The traps were checked and emptied weekly, and trapped moths were then sorted to identify FAW.

Field scouting was conducted at least twice a week, from the seedling and early whorl stages of the maize crop. This was also the time that farmers and extension workers sampled for egg masses, larvae, damage symptoms and the presence of other pests such as the African Armyworm (AAW) *Spodoptera exempta* (Walker) and stemborers. The maize fields were scouted using a "W" pattern approach, which involved sampling 10 consecutive plants at five different spots along the "W" transect (20). FAW and non-target moth counts from pheromone traps, and field scouting data were recorded and entered into the FAMEWS app. The mean percentage of plants infested with FAW, AAW, and stemborers, was automatically tabulated. Data were also collected on the condition of traps and whether the lures (pheromones) had been replaced. Additional data collected with the FAMEWS app included dates, country, geolocation, crop information (variety, planting date, irrigated or rain-fed, fertilizers used, and crop growth stage), general health of the crop, the cropping system (e.g. mono/intercropping, seasonal, rotation and push-pull), pest management practice adopted (chemical pesticides or biopesticides), and rainfall. Push-pull encompasses intercropping maize with the legume *Desmodium* spp. (*Desmodium intortum* for climate-smart push-pull or *Desmodium uncinatum*

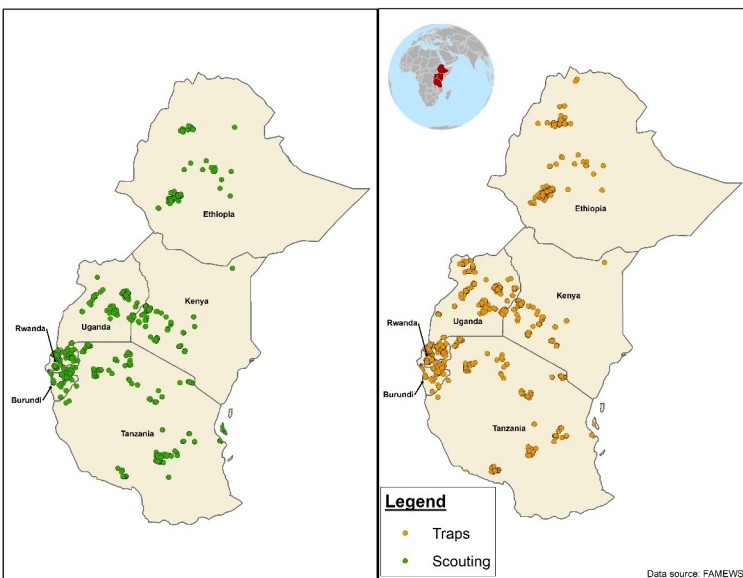

**Fig 2. Fall armyworm sample locations of the installed traps for adults and scouted farms for larvae in six eastern African countries.**

for conventional push-pull) and a border row of Napier grass *Pennisetum purpureum* or *Brachiaria brizantha* cv Mulato II (for climate-smart push-pull) around the plot; both *Desmodium* sp. and Napier grass are perennial fodder plants [18]. Seasonal cropping is a farming practice in which the same crop is grown in the same area for a number of growing seasons. In the case of maize, it is mainly rain fed, and the land remain fallow between seasons. While crop rotation is the practice of growing a series of different types of crops in the same area across seasons.

## 2.4. FAW monitoring data validation and processing

A national coordinator in each country was appointed to validate the data collected by the CFPs through FAMEWS, and who uploaded to the global platform maintained by FAO (http://www.fao.org/fall-armyworm/monitoring-tools/famews-global-platform/en/) through the global coordinator. The data presented in this study were downloaded from the FAO global platform, which has the entire database collected for the FAMEWS app between January 2018 to June 2019. The various data entries were officially requested from the FAO, cleaned and analyzed in Microsoft Excel and R software 3.6.1 [27].

## 2.5. Weather data and mapping methodology

Monthly rainfall data for 2018 and 2019 were sourced from Climate Hazards Group Infrared Precipitation with Station data (CHIRPS). CHIRPS incorporates satellite imagery with in-situ station data to create 1 km resolution gridded rainfall time series data (www.chc.ucsb.edu/data/chirps/). The geo-referenced points were used to extract monthly rainfall records (proxy data) in millimeters of the FAW infested areas in the six countries using the point sampling tool in QGIS software (http://qgis.osgeo.org). The FAW density was plotted against monthly rainfall data to establish whether there was a relationship between rainfall patterns and the abundance of FAW in the six countries. We interpreted effects of rainfall from a pest management perspective, whereby downpours, for instance could influence the pest flight and neonate movement and feeding on maize plants [28, 29]. Rainfall can both negatively and positively affect the general plant health, particularly in rain-fed cropping systems.

Spatial distribution of FAW density (number of insects per 1 km$^2$) in target countries, based on both trap and scouting data, were represented using heat maps. The FAW traps and scouting georeferenced data dating from January 2018 to June 2019 were organized on a quarterly basis based on the collection date. The points were clustered quarterly for the year 2018 to the 2$^{nd}$ quarter of 2019. The heat maps were developed using the kernel density tool in QGIS software (http://qgis.osgeo.org). Kernel density estimation is a powerful non-parametric technique for estimating probability density function of variables. The tool calculates the density of point features around each output raster cell. The density was calculated based on an accumulation of the number of FAW geo-referenced records in a sampled location, with a higher number of FAW records resulting in a higher value in the heat map. The heat maps were developed with the assumption that the sampling protocol for scouting and installation of traps was spatially unbiased. This analysis helps in identifying the hot-spot areas with high infestation rates. On the other hand, we utilized a maize cropping calendar which was freely available from FAO Global Information and Early Warning System (GIEWS: http://www.fao.org/giews/countrybrief/index.jsp) to explore the influence of maize sowing, growing, harvesting and fallow periods on FAW adults density during the main and second seasons. For a regional overview of FAW population dynamics, we also utilized FAW occurrence and density data from 11 African countries (six in East Africa and five in other regions) that are available in the FAO global platform. Further, data collected from pheromone traps (universal bucket traps) in the eastern and other African countries were quarterly mapped at the regional level.

Specifically, we overlaid the quarter FAW density heat maps in 2018 and 2019 to show the change in FAW density over time in each location, and differences in FAW density between locations at a point in time. Further, we mapped the all sites in the region where FAW was consistently observed across all the quarters in the year (i.e. year-round distribution).

## 2.6. Statistical analysis

Analyses were performed using R version 3.6.1 [27]. Trap counts were modelled using the Negative Binomial Model (NBM) which accommodates overdispersion of integer counts data. The NBM was used to evaluate the effect of main crop types, rotation crop and cropping systems on FAW trap counts. Incident rate ratios (IRR) were estimated for the different levels of each factor relative to a chosen reference level of a factor in question. IRR is a relative measure of incidence rate, such that IRR = 1 means no effect of the exposure, IRR > 1 means positive effect of exposure and IRR < 1 means negative effect of exposure. Similary, the NBM was also used to study the effect of crop phenology on trap counts.

For main crop factor, maize was used as a reference category while for cropping systems, we used seasonal cropping as a reference level. "Beans" was used a reference group for the crop rotation factor. The effect of maize crop stage on FAW trap counts and larval counts (as proportion of infested plants) was analysed for each country separately in view of high variability between countries. While the FAW trap counts were analysed using the NBM, proportion of larval infested plants were modeled using quasi-binomial model. All tests were performed at the 5% significance level.

## 3. Results

### 3.1. Crop diversity and cropping systems in East Africa

Maize and sorghum were the major crops (91.1% of all crops) in the target districts (n = 4165). Maize alone was a major staple food crop, representing 83.1% of all crops in the target districts. Other crops cultivated included beans, rice and millet, each representing less than 2% (Fig 3).

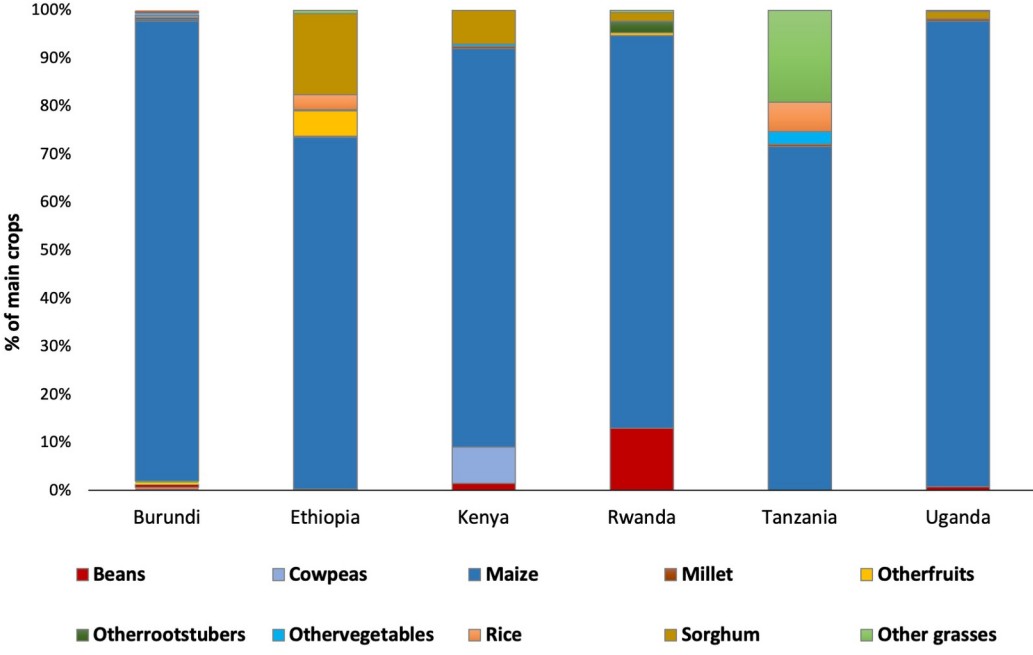

**Fig 3. Country-wise proportion of the main crop in the study districts.**

**Table 1. Regression parameter estimates and incident rate ratios (IRR) estimated from the negative binomial model on trap counts for the main crop factor adjusted for crop phenology with main crop 'Maize' used as a reference level[†].**

| Main crop | Estimate | Significant codes | IRR | 95% C.I for IRR |
|---|---|---|---|---|
| Intercept | 1.261 | | | |
| Barley | -1.517 | ns | 0.219 | (0.049, 1.845) |
| **Beans** | **-0.607** | * | **0.545** | **(0.338, 0.918)** |
| **Cowpeas** | **-2.922** | *** | **0.054** | **(0.018,0.186)** |
| Millet | 1.006 | ns | 2.733 | (0.636, 39.827) |
| Other cereals | -1.156 | ns | 0.315 | (0.062, 4.807) |
| **Other fruits** | **-2.148** | *** | **0.117** | **(0.073, 0.190)** |
| **Other grasses** | **-1.153** | *** | **0.316** | **(0.201, 0.517)** |
| Other roots & tubers | -0.023 | ns | 0.977 | (0.381, 3.507) |
| **Other vegetables** | **-1.906** | *** | **0.149** | **(0.054, 0.516)** |
| **Rice** | **-0.714** | ** | **0.480** | **(0.325, 0.775)** |
| **Sorghum** | **-1.471** | *** | **0.230** | **(0.180, 0.295)** |
| Soybean | -1.562 | ns | 0.210 | (0.022, 19.545) |
| Teff | -1.361 | ns | 0.257 | (0.042, 7.199) |
| Weeds | -0.764 | ns | 0.466 | (0.112, 4.792) |
| Wheat | -1.475 | ns | 0.229 | (0.026, 21.099) |
| Maize | Reference | | 1 | |

[†]Analysis based on sample size, n = 3438. IRR—Incident Rate Ratio is the exponentiated coefficient estimate. IRR = 1 means no effect of the exposure, IRR > 1 means positive effect of exposure, IRR < 1 means negative effect of exposure.

The diversity of main crops in the CBFAMFEW countries significantly affected FAW counts, Likelihood Ratio (LR) $\chi^2$ = 212.75, df = 15, p < 0.001. FAW trap counts were significantly lower in farms where the main crops were cowpeas, fruit trees, other grasses, vegetables, rice, and sorghum, relative to maize (Table 1).

Seasonal and rotation cropping systems were the dominant practices in the targeted districts of the six countries (n = 5291). Intercropping was observed in only 10% of the sampled sites. In Ethiopia, Rwanda, Tanzania and Kenya, intercropping is practiced mainly with beans and to a lesser extent with sorghum. Other crops included root and tuber crops, particularly in Rwanda, while in Tanzania other grasses (Napier, Panicum, and Kikuyu grass) are used as intercrops (Fig 4). The push-pull technology represented less than 1% of all cropping systems (Fig 4). Seasonal cropping system was the dominant practice in the districts surveyed in Kenya (43%; n = 417), Tanzania (74.86%; n = 871), Uganda (55.19%; n = 578), and Ethiopia (72.1%; n = 2535). On the other hand, crop rotation was mostly practiced in Burundi (60.8%; n = 301) and Rwanda (94.23%; n = 589) (Fig 4).

Higher numbers of moths were found under intercropping (2x) and crop rotation systems (3x) relative to the seasonal cropping system (Table 2). Moth numbers from the push-pull technology system were similar to those from the seasonal cropping system (Table 2).

Over 15 different types of crops were used for rotation, however; rotation with beans was the most commonly practiced in all countries. Maize followed by maize was widely practiced in Rwanda and Tanzania. Crop rotation with millet was widely practiced in Ethiopia, root and tuber crops were mainly used as rotation crops in Burundi, while cassava/manioc was used in Uganda (Fig 5). The numbers of FAW moths varied with the rotation crop (LR $\chi^2$ = 228.43, df = 16, p < 0.001). Rotation with barley, millet, root tubers, sorghum and soybean reduced the risk of FAW infestation relative to beans while cowpea increased the risk by 2.5-times relative to the reference crop, beans (Table 3).

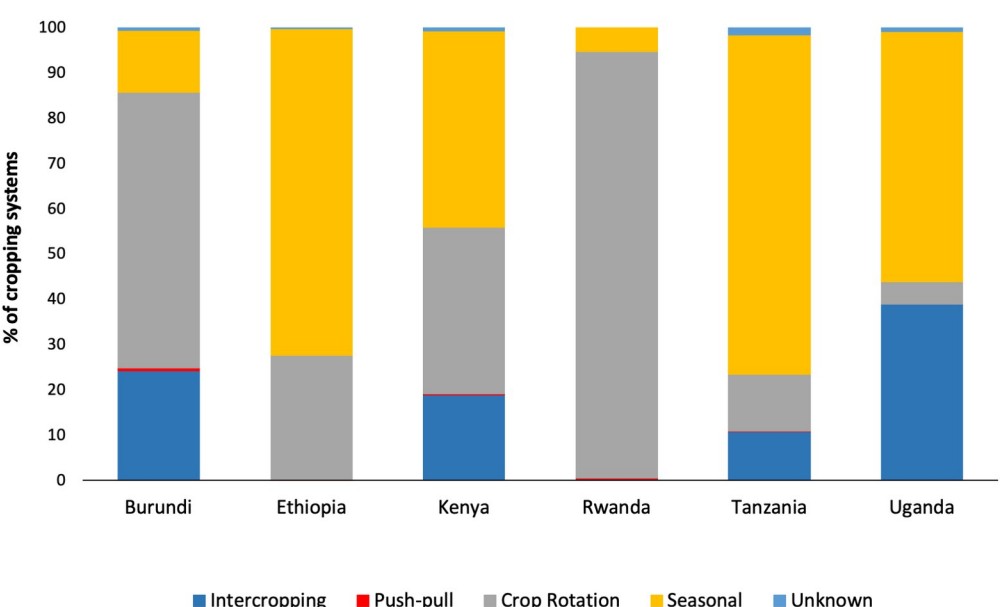

**Fig 4. Proportion of different cropping systems in East Africa.**

## 3.2. Use of pesticides

Extensive use of pesticides was reported in Tanzania, while the lowest use of pesticides was reported in Ethiopia. During the surveyed period for the target sites, the quantity of biopesticides used in FAW management was very low (307 liters) as compared with chemical pesticides (10,293 liters) (Fig 6A). The most commonly used chemical pesticides that were applied by farmers in the East African region for FAW control were alpha-cypermethrin (Pyrethroid), chlorpyriphos and malathion (organophosphates), and lufenuron (benzoylurea insecticide) (Fig 6B). Chlorpyrifos was the most commonly used in Ethiopia. Chlorpyrifos was also used in Tanzania, Uganda and Burundi. Lufenuron was mainly used in Kenya and Tanzania (Fig 6B). Trap and larval counts were highly variable among chemical pesticides and countries. High adult and larval counts were found in Burundi, Kenya and Rwanda in locations where alpha-cypermethrin was used. Relatively lower adult and larval counts were recorded where chlorpyriphos was used except in Tanzania, Ethiopia and Uganda, where larval counts were high. There was lower trap and larval counts in all countries where malathion was used. In areas where lufenuron was used, high larval counts were recorded in Kenya, whereas in Tanzania both trap and larval counts were low in areas where lufenuron was applied.

**Table 2. Regression parameter estimates and incident rate ratios (IRR) estimated from the negative binomial model on trap counts for the cropping system factor adjusted for crop phenology with 'Seasonal cropping' used as a reference level[†].**

| Cropping System | Estimate | Significant codes | IRR | 95% C.I for IRR |
|---|---|---|---|---|
| Intercept | 0.552 | | | |
| **Intercropping** | **0.849** | *** | **2.336** | **(1.839, 3.006)** |
| Push-pull | -0.010 | ns | 0.990 | (0.210, 14.276) |
| **Rotation** | **1.149** | *** | **3.154** | **(2.740, 3.634)** |
| Seasonal cropping | Reference | | 1 | |

[†]Analysis based on sample size, n = 3423. IRR—Incident Rate Ratio is the exponentiated coefficient estimate. IRR = 1 means no effect of the exposure, IRR > 1 means positive effect of exposure, IRR < 1 means negative effect of exposure.

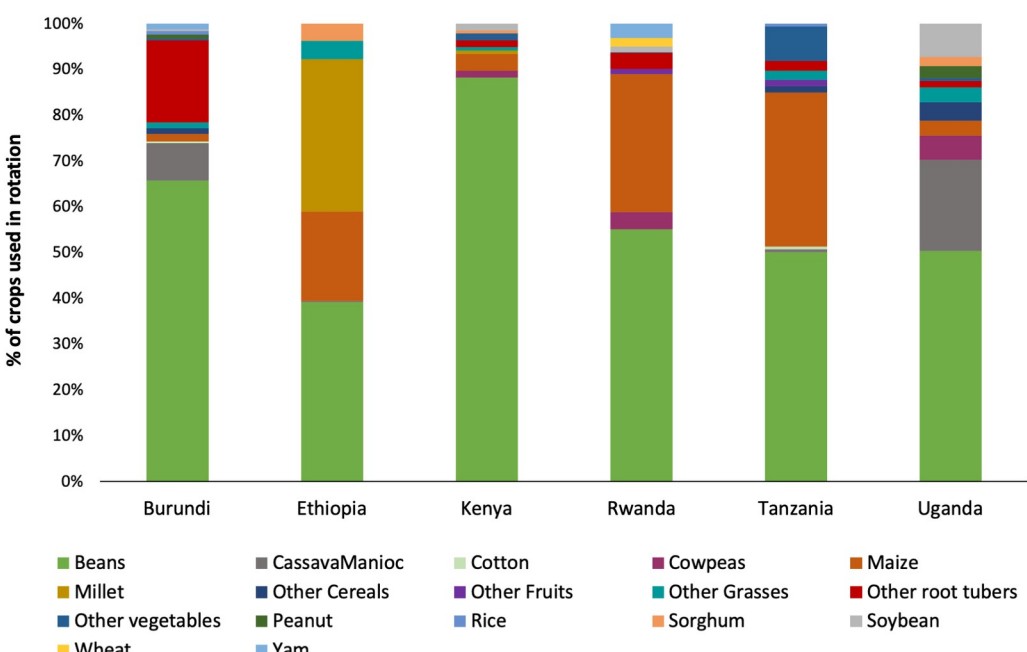

**Fig 5. Proportion of crops used in rotation cropping system in the study districts of six eastern African countries.**

The biopesticides that were applied included NPV (nuclear polyhedrosis virus), *Bacillus thuringiensis* (Bt), *Metarhizium*, *Beauveria* and spinosad. *Metarhizium anisopliae* was the most commonly used biopesticide in the region, except in Burundi, where NPV was used.

**Table 3. Regression parameter estimates and incident rate ratios (IRR) estimated from the negative binomial model on trap counts for the rotation crop factor adjusted for crop phenology with beans crop was used as a reference level[†].**

| Rotation crop | Estimate | Significant codes | [+]IRR | 95% C.I for IRR |
|---|---|---|---|---|
| Intercept | 1.966 | | | |
| **Barley** | **-2.109** | *** | **0.121** | **(0.045, 0.390)** |
| Cassava Manioc | -0.533 | ns | 0.587 | (0.339,1.096) |
| **Cowpeas** | **0.940** | ** | **2.559** | **(1.378, 5.453)** |
| Maize | -0.031 | ns | 0.969 | (0.753, 1.258) |
| **Millet** | **-3.150** | *** | **0.043** | **(0.028, 0.066)** |
| Other cereals | -0.191 | ns | 0.826 | (0.281, 3.982) |
| Other fruits | 1.013 | ns | 2.755 | 0.887, 15.150) |
| Other grasses | -0.449 | ns | 0.638 | (0.054, 776.409) |
| **Other roots & tubers** | **-0.999** | *** | **0.368** | **(0.230, 0.622)** |
| Other vegetables | 0.313 | ns | 1.367 | (0.610, 3.888) |
| Peanut | -2.098 | ns | 0.123 | (0.027, 1.099) |
| Rice | -0.402 | ns | 0.669 | (0.058, 812.929) |
| **Sorghum** | **-2.396** | *** | **0.091** | **(0.038, 0.249)** |
| **Soybean** | **-1.206** | ** | **0.299** | **(0.137, 0.787)** |
| Wheat | -1.143 | ns | 0.319 | (0.109, 1.409) |
| Yam | -0.749 | ns | 0.473 | (0.216, 1.268) |
| Beans | Reference | | 1 | |

[†]Analysis based on sample size, n = 1479. IRR—Incident Rate Ratio is the exponentiated coefficient estimate. IRR = 1 means no effect of the exposure, IRR > 1 means positive effect of exposure, IRR < 1 means negative effect of exposure.

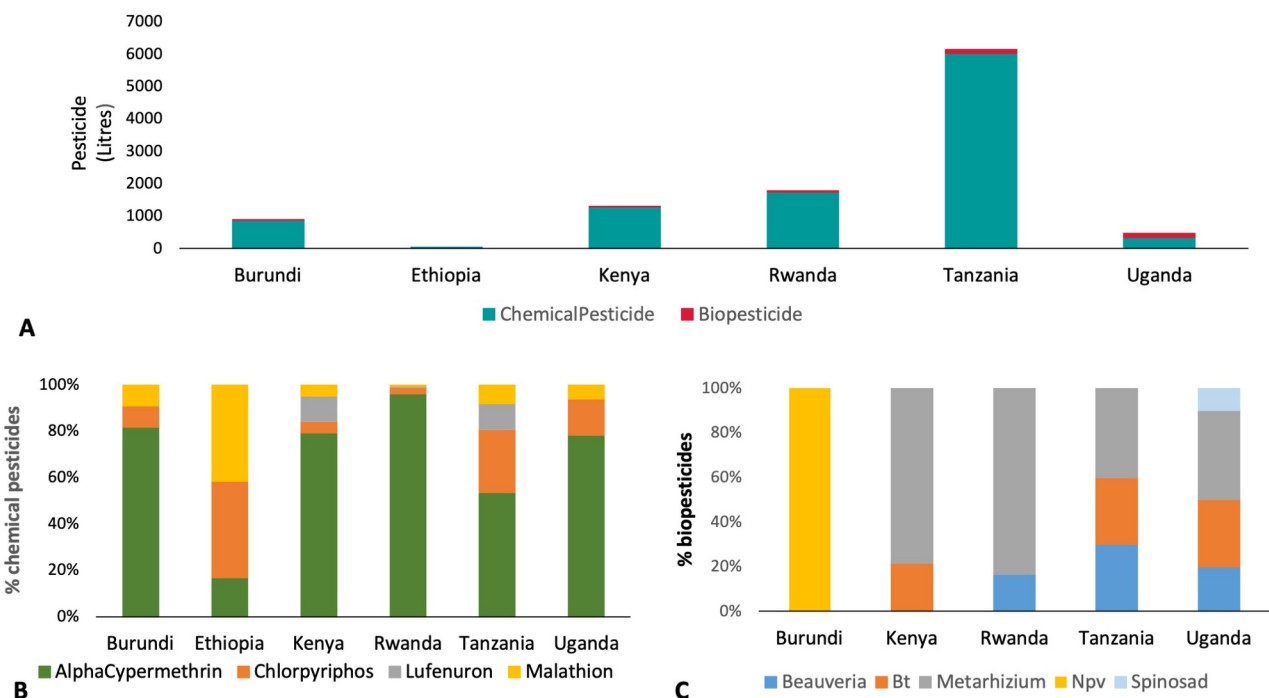

**Fig 6. Quantity of different groups of pesticide use in fall armyworm management in eastern African countries (accumulating the quantity of all the pesticides that were recorded in liters for the six countries, individually, both for chemicals and biopesticides).**

*Metarhizium anisopliae* was commonly used in Rwanda, Kenya, Uganda, and Tanzania. *Beauveria bassiana* was also used in Tanzania, Rwanda and Uganda, while Bt was applied in Tanzania, Uganda and Kenya. Spinosad was only recorded in Uganda (Fig 6C).

### 3.3. Fall armyworm dynamics and rainfall

FAW infestation was low when rainfall was at its peak, and then increased when rainfall subsided. This trend was observed in Ethiopia, Kenya, Uganda and Rwanda. In Tanzania and Burundi, however, that trend was not clear.

In Ethiopia, two sharp increases in FAW captures were observed. The first increase occurred in September, coinciding with the mid-growing phase of the Meher (long rains) season and the harvest phase of the Belg (short rains) season. The trap captures declined towards December. The adult population started to build up again towards the beginning of the Belg and Meher seasons (Fig 7A). In Kenya, two peaks of FAW abundance were noted, coinciding with the two cropping seasons. These peaks appeared when rainfall was low. The first FAW peak was observed between July and August, coinciding with the initiation of the growing phase of season 1. A second peak was observed in February–March, coinciding with a low rainfall period and the harvesting time and beginning of season 2 (Fig 7B).

In Uganda, very low numbers of adult moths were found and there was one peak of larval infestation noted from September (Fig 7C). In Rwanda and Burundi, the peak increase in FAW abundance started between November and December (Fig 7D and 7E), and although larval abundance declined in both countries, moth numbers increased in Rwanda (Fig 7E). In Tanzania, an increase in larval infestation started in December and ended in April (Fig 7F).

Correlation analysis between larval and adult counts and rainfall showed that in Ethiopia, the number FAW moth was not significantly correlated with rainfall ($r = -0.18$; $p = 0.5$) and

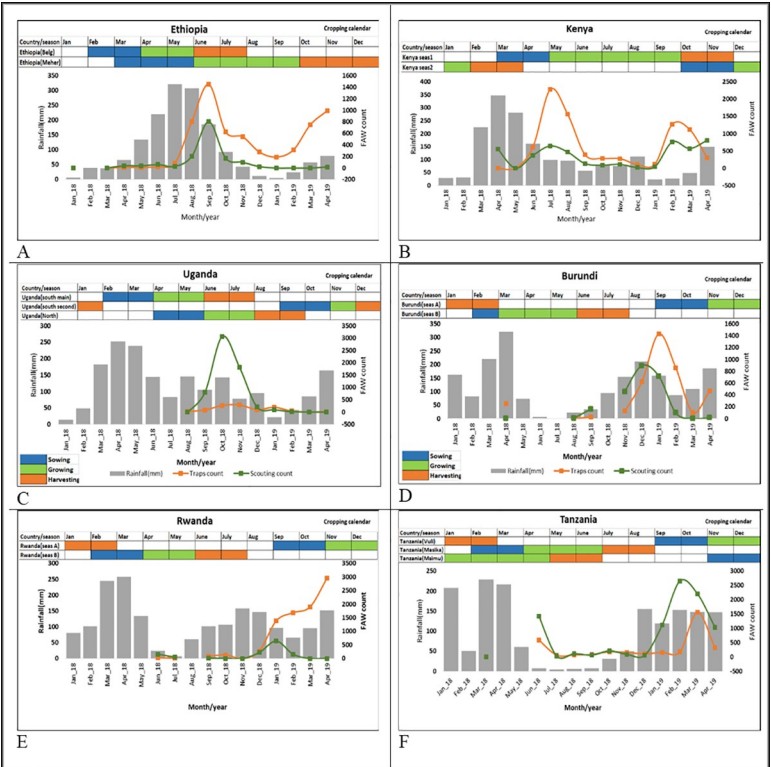

**Fig 7. Relationship between rainfall pattern, cultural calendar and fall armyworm trap captures and field incidence in East Africa.**

similarly, FAW larval counts and rainfall were not correlated (r = 0.01; p = 0.9). This was true for all the other five countries; Kenya, Uganda, Tanzania, Rwanda and Burundi.

## 3.4. Adult trap counts and larval scouting of fall armyworm in relation to maize crop phenology

At the sowing and seedling stage (grey and umber bars), the trap count (n = 5550) and the scouting count (n = 5846) of FAW were low. As the crop matures to the vegetative stage (green bar), a sharp build-up of FAW adults (trap count, n = 42,643) and larvae (scouting, n = 66,221) was recorded. This was followed by a progressive decline in trap catches (n = 28,591) and the number of larvae per plant (n = 10,065) as the crop reached the reproductive and maturity stages (blue and brown bars). It is important to note a sharper decrease in larvae counts (scouting), as compared with adult catches (trap count), when the crop approached maturity (Fig 8). A country by country analysis for FAW moth counts occurring at different maize crop phenologies suggested significant differences among stages in all countries except Rwanda, which had similar numbers for all stages (Table 4A). Most countries recorded high moth numbers for the vegetative and reproductive stages, although numbers were relatively low in Ethiopia and Tanzania for all stages. Uganda had high numbers of moths during the sowing stage and then low number for the rest of the crop cycle. For infested plants, all stages produced similar percentages in Kenya and Rwanda (Table 4B). Lower numbers of infested plants were found during the maturity stage in Burundi, Ethiopia, and Uganda; Tanzania had the lowest percent infested plants at the seeding stage.

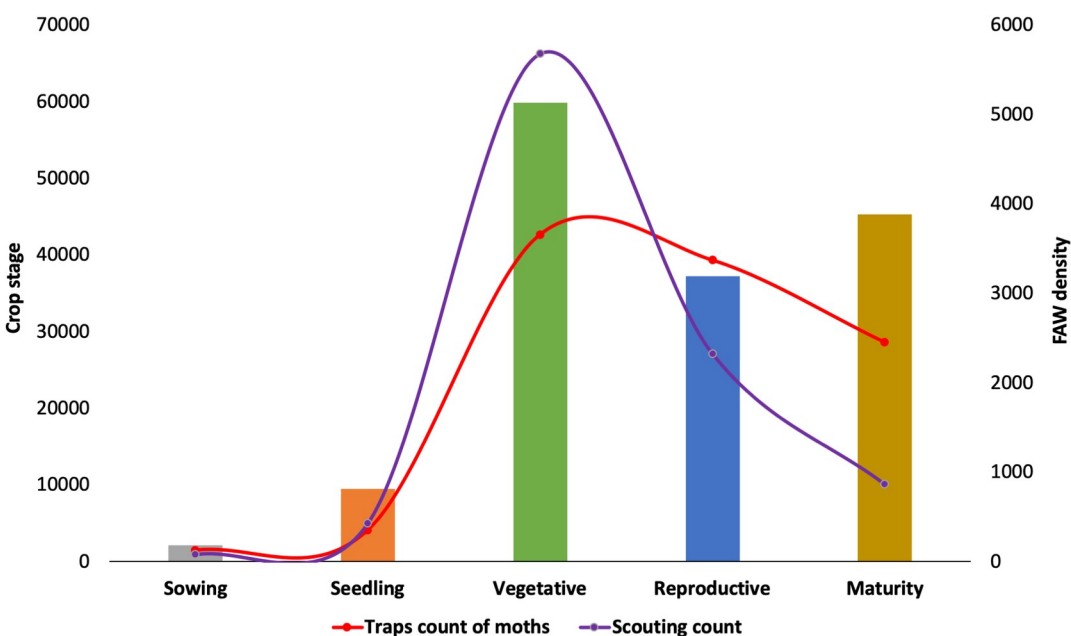

**Fig 8. Changes in adult captures (trap count) and larval population (field scouting) of fall armyworm in different phenology stages of the crop in East Africa.**

## 3.5. Spatial and temporal infestation of fall armyworm

The mapping of total FAW adults and larvae counts collected, indicated a clear progression over time of the infestation across East Africa (Fig 9). The figures in the legend represent the

**Table 4. Mean (±SE) adult FAW abundance (A) and mean (±SE) percentage FAW larva infested plants (Scounted larva) (B) for each country across maize crop stage and the corresponding test statistics for the Negative binomial model for adult FAW abundance and for the quasi-binomial model for percentage larva infested plants.**

| A. Adult FAW counts | | | | | | | |
|---|---|---|---|---|---|---|---|
| | | | Maize crop stage | | | | |
| Country | Sowing | Seedling | Vegetative | Reproductive | Maturity | χ2, df = 4 | P-value |
| Burundi | 0.0 | 17.2 ± 11.5 | 9.4 ± 1.4 | 33.8 ± 10.1 | 12.2 ± 6.9 | 38.37 | <0.0001 |
| Ethopia | 4.5 ± 4.2 | 3.1 ± 0.2 | 4.5 ± 1.2 | 2.7 ± 0.4 | 1.4 ± 0.1 | 37.01 | <0.0001 |
| Kenya | 1.5 ± 1.5 | 18.0 ± 1.1 | 23.7 ± 3.8 | 30.7 ± 7.2 | 13.6 ± 4.5 | 14.56 | 0.006 |
| Rwanda | 12.3 ± 1.4 | 14.1 ± 1.3 | 21.7 ± 8.7 | 13.1 ± 2.4 | 12.3 ± 1.4 | 7.40 | 0.118 |
| Tanzania | 1.6 ± 0.6 | 4.6 ± 1.1 | 4.1 ± 0.6 | 4.7 ± 0.6 | 3.8 ± 0.4 | 13.50 | 0.009 |
| Uganda | 34.7 ± 25.8 | 3.2 ± 0.7 | 1.6 ± 0.2 | 1.4 ± 0.2 | 1.4 ± 0.4 | 27.70 | <0.0001 |
| **Mean** | **5.3 ± 1.7** | **8.0 ± 0.7** | **7.4 ± 0.8** | **11.4 ± 1.7** | **3.7 ± 0.3** | | |
| **B. Percentage FAW larva infested plants** | | | | | | | |
| | | | Maize crop stage | | | | |
| Country | | Seedling | Vegetative | Reproductive | Maturity | **F-value** | P-value |
| Burundi | | 14.0 ± 7.8 | 22.1 ± 1.6 | 12.7 ± 2.3 | 3.8 ± 1.3 | $F_{3,198}$ = 9.62 | <0.0001 |
| Ethopia | | 17.0 ± 1.7 | 14.5 ± 0.9 | 20.1 ± 0.9 | 11.3 ± 0.2 | $F_{3,85}$ = 3.08 | 0.032 |
| Kenya | | 10.0 | 38.7 ± 3.0 | 34.0 ± 2.2 | 21.1 ± 2.7 | $F_{3,81}$ = 2.54 | 0.062 |
| Rwanda | | 0.8 ± 0.3 | 12.9 ±3.0 | 7.9 ± 1.7 | 3.7 ± 0.7 | $F_{3,146}$ = 1.84 | 0.142 |
| Tanzania | | 14.3 ± 1.8 | 29.5 ± 1.8 | 20.3 ± 2.0 | 23.4 ± 1.5 | $F_{3,174}$ = 6.80 | <0.001 |
| Uganda | | 31.4 ± 4.6 | 46.8 ± 2.3 | 39.7 ± 3.0 | 7.1 ± 2.2 | $F_{3,242}$ = 12.3 | <0.0001 |
| **Mean** | | **7.7 ± 0.8** | **30.6 ±1.0** | **23.3 ± 0.9** | **9.5 ± 0.4** | | |

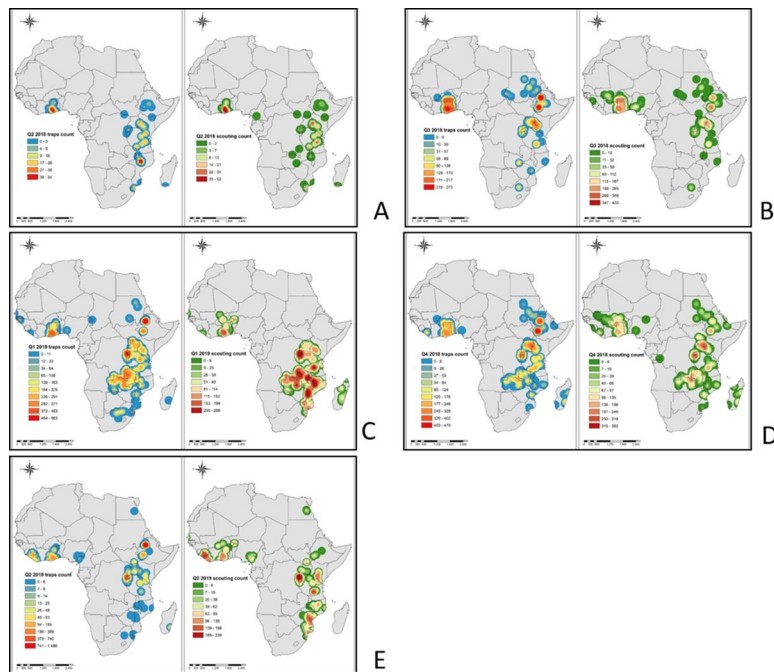

**Fig 9. Evolution of fall armyworm infestation according to the time for scouting and trap counts.** A) The second quarter of 2018, i.e. April, May and June; B) The third quarter of 2018, i.e. July, August and September; C) The fourth quarter of 2018, i.e. October, November and December; D) The first quarter of 2019, i.e. January, February and March; E) The second quarter of 2019, i.e. April, May and June.

total number of adults and larvae collected respectively on a quartely basis. The maps present a constant FAW adults and larvae presence, especially in irrigation and lake areas in the main and second cropping seasons. Regarding the onset of the maize season, regions of initial infestations or population build-up were noted around the humid zones of Lake Tanganyika in Tanzania, around Lake Victoria in Uganda, Tanzania and Kenya, and around Lake Malawi in Tanzania and Mozambique (second quarter of 2018, i.e. April, May and June). The activity of adult FAW was considerably high in the Lake Malawi basin of Mozambique, while larval infestation was high in the Lake Victoria basin and the Lake Malawi basin in Mozambique (Fig 9A). As the maize season progressed in the different countries in the third quarter of 2018 (July–September), infestations expanded towards the north, with considerable increases in adult and larval activity around the Lake Victoria basin in Kenya and Uganda and the Lake Kivu and Tanganyika basins in Rwanda and Burundi, and Ethiopia, Southern Sudan and Sudan (Fig 9B). In the fourth quarter of 2018, i.e. October, November and December, FAW adult and larval population expanded further in Rwanda, Burundi, Kenya, Uganda and Ethiopia. The FAW population build-up expanded extensively in Zambia, Mozambique and Madagascar (Fig 9C). During the first quarter of 2019, i.e. January, February and March, the severe infestation of FAW (both adult captures in traps and larval counts) was observed in most regions of East and southern Africa, apart from South Africa, Botswana and Namibia (Fig 9D). In the second quarter of 2019, i.e. April, May and June, there was a substantial decline in FAW population in all countries and this remained confined to most humid regions, as observed in the second quarter of 2018 (Fig 9E). The map similarities between FAW adult catches and larval infestations are noteworthy. Figs 10 and 11 show FAW adults progression according to seasonal maize cropping calendars in the six East African countries targeted. The heat maps indicate the changes in the intensity of the trap counts during the sowing, growing, harvesting

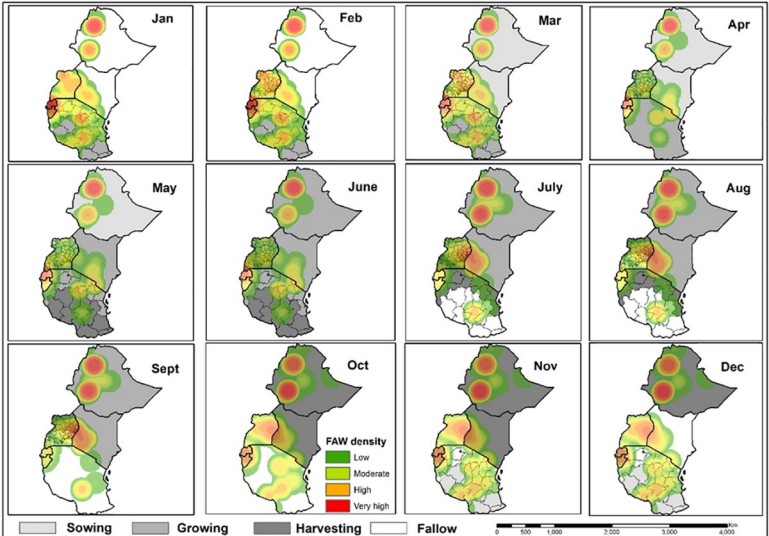

**Fig 10. Maize cropping main seasons and fall armyworm trap counts dynamics in eastern Africa.**

and fallow periods across the six countries. The combined map of the year-round FAW occurrence reveals remarkable hotspots with high FAW adults prevalence (Fig 12).

## 4. Discussion

### 4.1. Crop diversity and cropping systems

Cereal production in eastern and southern Africa is dominated by maize (70%) with sorghum accounting for 7% and millet 2% of total cereal produced [30]. Except for millet and root tubers, all main crops had a negative influence on FAW trap counts, including sorghum, Africa's second most important cereal in terms of tonnage [31]. Data suggest more attention is needed on other crops attacked by FAW such as sorghum [32], Africa's second most important cereal in terms of tonnage [31].

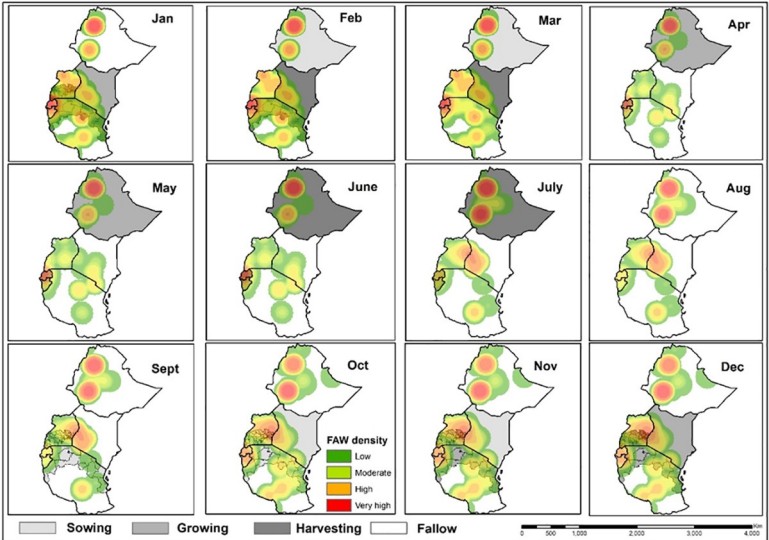

**Fig 11. Maize cropping second seasons and fall armyworm trap counts dynamics in eastern Africa.**

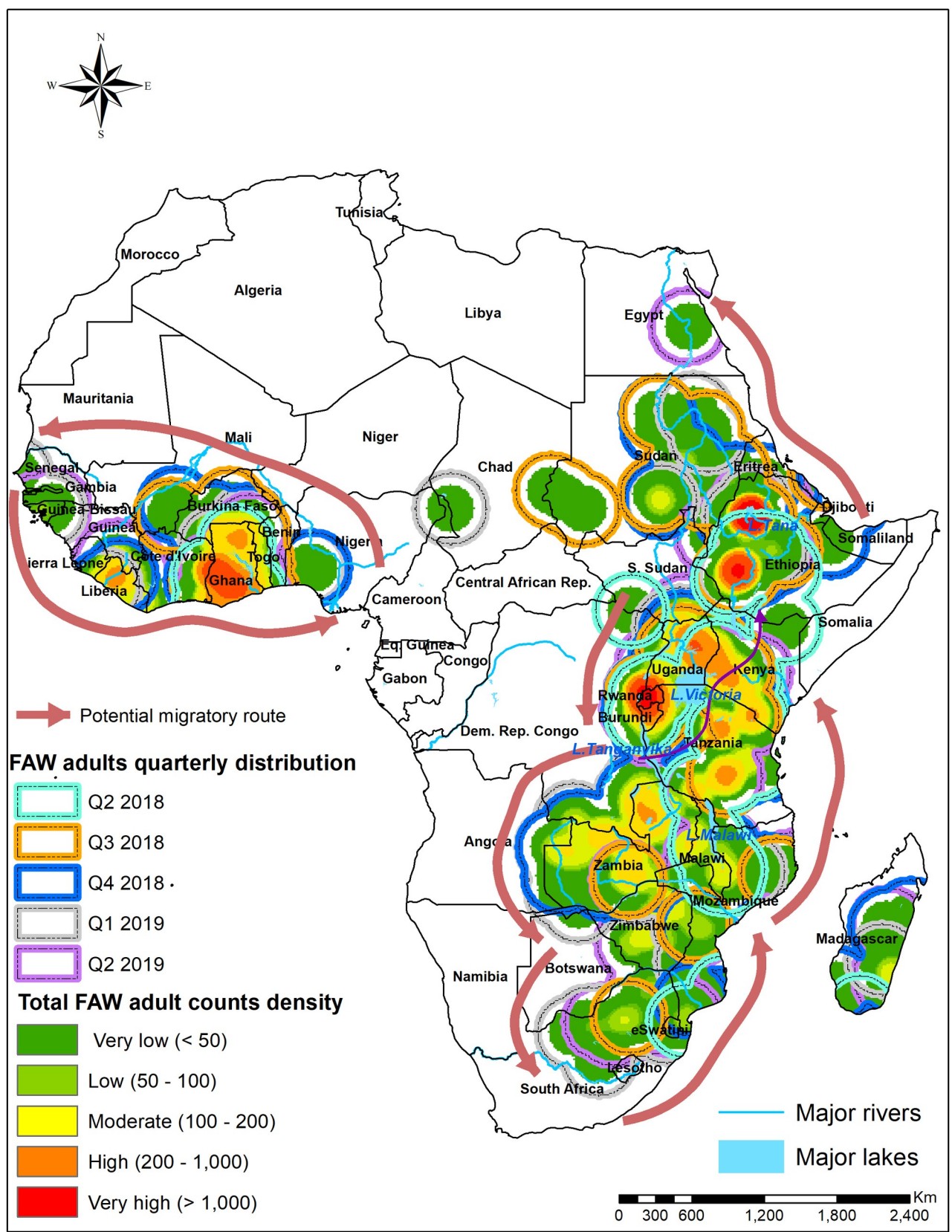

**Fig 12. FAW adults spatial distribution consistently observed throughout the years in 2018 to 2019.**

The study found that the type of cropping system varied between countries. Seasonal cropping systems were the dominant practices in Ethiopia, Kenya, Tanzania and Uganda, while in Burundi and Rwanda, crop rotation was the common practice. The variation in cultural practices can be linked to country sizes and land availability. Shortage of land is indeed a reality in Rwanda, and this has resulted in a cascade of policies including the "Land Consolidation and Crop Intensification Programmes", which might favor or mitigate the pest [33]. Our study demonstrated that cropping systems play an essential role in FAW population dynamics as they define the availability of alternative host plants that can sustain the pest over time [20, 34].

Seasonal cropping suggested FAW infested crops in-season in certain regions and there was no continuous presence of host plants to attack. FAW trap counts were lower with seasonal cropping systems compared with crop rotation and intercropping systems.

Our study revealed that a wide range of crops was used in rotation with maize in East Africa. Beans were the most commonly used for rotation. Planting of maize followed by maize in the subsequent season is expected to sustain the pest year round and this practice was observed in Ethiopia, Rwanda and Tanzania. Crop rotation and intercropping are disruptive IPM strategies and may either break the cycle of the pest or deter feeding or oviposition through the release of a volatile complex, or serve as a physical barrier to the progression of neonates [34, 35]. A perfect example of that type of cropping system is push-pull technology, which reduced both adult and larval counts. Midega et al. [20] observed reductions of 82.7% in the average number of FAW larvae per plant and 86.7% in plant damage per plot while applying climate-adapted push-pull technology, as compared with monocropped maize. Similarly, Hailu et al. [36] reported that intercropping of maize with leguminous crops provided significant reduction of stemborer and FAW incidence, as compared with monocropped maize. However, a previous study conducted in Zimbabwe showed that intercropping maize with pumpkin increased FAW damage [34]. In our study, intercropping represented 10% of the total survey whereas Push-Pull technology was only 1%; suggesting that more effort is needed to scale out these practices.

Over 15 crops were used in crop rotation and six had a reducing effect on adult FAW. Hence, there is need to validate the role of other crops through chemical ecology and investigate plant to plant communication, volatile release to attract natural enemies and multi-trophic interactions [16].

## 4.2. Use of pesticides and biopesticides

The efficacy of chemical pesticides in Africa is questionable as reported in recent studies in Kenya, Ethiopia and Zimbabwe [34, 37]. The indiscriminate use of chemical pesticides is not a sustainable way of managing this pest, which has shown resistance towards many compounds in its native region [23, 38]. In the Americas, FAW resistance to mode-of-action categories 1A (Carbamates), 1B (Organophosphates), and 3A (Pyrethroids-Pyrethrins) has been reported [15, 22, 23]. Furthermore, Africa is a home to nine species of *Spodoptera*; hence, it is crucial to consider natural enemies such as *Telenomus remus* Nixon (Hymenoptera: Platygastridae) [39], *Cotesia icipe* Fernández-Triana & Fiaboe (Hymenoptera: Braconidae), *Chelonus curvimaculatus* Cameron (Hymenoptera: Braconidae), *Charops ater* Szépligeti, *Palexorista zonata* (Curran) and *Coccygidium luteum* (Brullé) [40–42]. The heavy application of chemical pesticides adversely impact these beneficial organisms, which might otherwise reduce the levels of FAW infestation. Although the negative effects of insecticides haven't been tested against the natural enemies identified in Africa, studies from the Americas have shown these effects in related parasitoid and predator species [43, 44]. Moreover, these chemical pesticides (chlorpyriphos, malathion, alpha-cypermethrin and lufenuron) are associated with severe environmental impact

[45]. Where insecticide application cannot be avoided, the use of selective products that have reduced impact on non-target and beneficial organisms must be encouraged. This emphasizes the need for policy regulation and farmer training for the safe use and selection of pesticides in FAW management. The variability in chemical applications between Ethiopia, Kenya and Tanzania observed in our study may account for the variability in the diversity of natural enemies and differences in parasitism levels [40]. This calls for a paradigm shift in FAW management, especially in farmer training in the agroecological management of insect pests. The results of this study suggest that greater effort is needed on biological control (classical, inundative and augmentative methods) and agroecological approaches [16]. Other additional agroecological approaches which could aid in enhancing the efficacy of IPM tools include minimum tillage, biomass mulching, use of cover crops and alley cropping, agroforestry and natural habitats [16, 34].

The study also noted that on farms where biopesticides were used, percent infested plants was higher. This result is contrary to the mode of action of biopesticides and could be either due to the inefficacy of product used or little amount of data collected. Biopesticides used include fungal-based products as well as NPV. Private companies have been promoting biopesticides in the region. Generally, biopesticides are less toxic and sustainable [19], although they may negatively influence natural enemy populations [46]. However, biopesticide use needs to be promoted with the involvement of private sector companies to engage in production. Several potential biopesticide candidates have been identified in Africa. For instance, recent studies have revealed ovicidal effects of over 90% mortality-causing of already commercialized and other potential biopesticides ICIPE 7 and ICIPE 78 for management of FAW [47]. Our study, therefore, calls for the harmonization of regional policies in terms of validation and registration procedures for biopesticides that will ensure their accessibility, affordability and availability to cereal smallholder farmers.

## 4.3. Fall armyworm dynamics and rainfall

The impact of climate (temperature and rainfall) on pest pressure has been reported by several authors in the Americas. Heavy downpours are detrimental to FAW population build-up [28, 48, 49]. [48] demonstrated that egg dislodgement was more frequent in the rainy seasons than during in the dry seasons, indicating that rain and associated weather factors caused high rates of dislodgement. [49] stated that the wash-off of eggs and larvae due to the direct impact of rain, and the drowning of young larvae after rains, were considered to be the major causes of the disappearance of diamondback moth (Lepidoptera: Plutellidae) eggs and larvae. In our study, we had no evidence of significant correlation between rainfall and the FAW population. Accordingly [11], severe outbreaks of armyworms usually coincided with the onset of the wet season, mainly when the new cropping season followed a long period of drought.

We observed that the peak of FAW populations coincided with reduced rainfall and crop stage (planting, growing and harvesting) [50]. Integration of weather information in FAW management has been reported in South America [29]. More effort is therefore needed to determine the relationship between rainfall and FAW population fluctuations.

## 4.4. Dynamics of fall armyworm in relation to the phenology of the crop

FAW infestation was significantly influenced by crop phenology for both adults and larvae. Adult FAW density was significantly influenced by all maize stages, whereas plant infestation was mainly influenced by the reproductive and vegetative stages. Pheromone traps attract male moths regardless of the cereal crop stage available or the presence of alternative host plants. On the other hand, larval counts were only reported from maize. Since the FAW

lifecycle lasts 30 days under optimum conditions [51], 3–4 generations are expected within an entire maize cycle, assuming all farmers in a region plant at the same time. This is usually not the case, and considering spatial variations at the start of seasons within a country and delays in planting, crop diversity as well as variation in cropping systems and type of rotation crops, several staggered generations can be observed under optimal condition. This has been illustrated with FAMEWS data that reveals the building-up of FAW populations as the cropping season unfolds. Considering the various IPM strategies available (biopesticides, handpicking, trapping, egg crushing, parasitoids and natural enemies, push-pull, intercropping, etc.), farmers should be given options for the deployment of best-bet technologies at each growth stage of the crop. For instance, at the sowing and seedling stages where the adult and larval counts were low, an initial release of parasitoids coupled with egg crushing could prevent or suppress the build-up of FAW populations. At the vegetative stage where adults and immature stages are equally abundant, the release of parasitoids and application of egg-killing biopesticides could be appropriate. At the reproductive and maturity stages, predatory bugs that target mostly adults and pupae must be recommended. The expansion of Push-Pull technology requires vibrant local seed systems to ensure availability and accessibility of companion plants that are currently imported from abroad at $50/kg and $30 for *Desmodium* and *Brachiaria* respectively. It is expected that local seed production would bring the cost of seeds to $10. Although ICIPE 7 and ICIPE 78 have been identified for FAW, registration processes in East Africa are underway. Cultural practices such as handpicking and use of sand or ash are readily available to farmers at no cost, although with little known efficacy.

**4.4.1. Spatial and temporal infestation of fall armyworm.** Our study revealed spatial and temporal dynamics of the FAW population and the infestation cycle. Most East Africa countries experience two rainy seasons; September-January and February-March-June. In southern Africa (Zambia, Malawi and Zimbabwe), planting begins in November-December and harvesting takes place in March-April (Fig 7). This suggests that when infestation is at its peak, for instance in Q2 and Q3, FAW infestations might cover both eastern and southern Africa with overlapping generations. However, most southern African countries have unimodal rainfall pattern; therefore, FAW may occur around irrigation sites, e.g., Q3. Considering the variable rainfall pattern, maize seasonality, and variations in planting dates within countries and between countries and regions, there is a possibility of seasonal infestation in Africa compared to the Americas where FAW faces winter [52]. These variations imply potential FAW migratory routes within East Africa and between East and Southern Africa considering the long dry spell in southern African countries. Similarly, there is possibility of migration from Ethiopia, northwards to Sudan and Egypt following Nile river or Southwards during onset of maize season in Kenya. In West Africa, countries like Ghana and Cote d'Ivoire also experience major and minor seasons. The availability of alternative hosts and their flight capability might fuel a spatial expansion pattern at the regional level [11, 26, 53, 54] similar to *S. exempta* [55, 56]. Our study demonstrated that in addition to seasonal variabilities, there are areas with continuous farming through irrigation; for example, transboundary lake regions are permanent hotspots, from which FAW might spread to other regions when weather is favorable, and host plants/crops are available. From a strategic point, hotspot areas such as irrigation sites should be high priority for interventions, with a strong emphasis on biological control. In the context of preparedness, our study supports the need for further studies to test FAW migratory pattern through the use of additional information such radar, high-altitude sampling, back-track wind trajectory and genetic population characterization to determine source of infestation [57].

The piloting phase of the CBFAMFEW endeavor provides in-depth insight into, and trends of, FAW bioecology in Africa which is a prerequisite for any meaningful management

intervention. If rolled out, the CBFAMFEW model could provide more accurate insight into FAW ecology and biology, leading to greater agroecological approaches. Intercropping and augmentative and conservation biological control approaches should be undertaken, which would target off-seasons to foster the increase of populations of natural enemies that can, in turn, control the pest during the in-seasons. The integration of weather data could help to reduce unnecessary pesticide applications, and save costs for farmers and reduce heavy environmental hazards. However, more effort is needed to establish the link between rainfall and FAW occurrence. Cropping systems that contribute to the management of FAW, such as intercropping and push-pull technology, need to be scaled out at regional levels in tandem with strategies encouraging their adoption by farmers. The results of this study demonstrate that conditions that favor FAW establishment exist where studies were conducted. Although most countries have established FAW National Strategy and Task Force, the containment of the pest can only be achieved at regional and global levels, now that the pest has been reported in Asia [58, 59]. This study, therefore, recommends the continuous support of the CBFAM-FEW by African governments and the United Nations, the sharing of experience between countries, regions and continents, and the promotion of CBFAMFEW to allow for timely evidence-based decision-making processes for preparedness. The study also suggest further studies to test the suggested FAW migratory pattern through the use of additional information.

## Supporting information

**S1 Fig.**
(JPG)

## Acknowledgments

We thank FAO, which made the data used in this paper freely available. The authors express their gratitude to farmers, extension agents and all technical staff members involved in the data collection in the different countries. USAID/OFDA contributed to the national capacity that enabled the data to be generated.

## Author Contributions

**Conceptualization:** Saliou Niassy, Mawufe Komi Agbodzavu, Girma Hailu, Sevgan Subramanian.

**Data curation:** Saliou Niassy, Mawufe Komi Agbodzavu, Emily Kimathi, El Fatih M. Abdel-Rahman, Daisy Salifu.

**Formal analysis:** Mawufe Komi Agbodzavu, Emily Kimathi, El Fatih M. Abdel-Rahman, Daisy Salifu.

**Funding acquisition:** Yeneneh T. Belayneh, Sunday Ekesi, Sevgan Subramanian.

**Investigation:** Saliou Niassy, Berita Mutune, Girma Hailu, Elias Felege.

**Methodology:** Saliou Niassy, Emily Kimathi, El Fatih M. Abdel-Rahman, Daisy Salifu, Henri E. Z. Tonnang.

**Project administration:** Saliou Niassy, Berita Mutune, Girma Hailu, Elias Felege, Sevgan Subramanian.

**Resources:** Yeneneh T. Belayneh, Sunday Ekesi, Sevgan Subramanian.

**Software:** Mawufe Komi Agbodzavu, Emily Kimathi, El Fatih M. Abdel-Rahman, Daisy Salifu.

**Supervision:** Saliou Niassy, Yeneneh T. Belayneh, Elias Felege, Henri E. Z. Tonnang, Sunday Ekesi, Sevgan Subramanian.

**Validation:** Emily Kimathi, El Fatih M. Abdel-Rahman, Daisy Salifu, Henri E. Z. Tonnang.

**Visualization:** Saliou Niassy, Mawufe Komi Agbodzavu, Emily Kimathi, El Fatih M. Abdel-Rahman, Henri E. Z. Tonnang, Sevgan Subramanian.

**Writing – original draft:** Saliou Niassy, Mawufe Komi Agbodzavu, Emily Kimathi, Sevgan Subramanian.

**Writing – review & editing:** Saliou Niassy, Mawufe Komi Agbodzavu, Berita Mutune, El Fatih M. Abdel-Rahman, Daisy Salifu, Girma Hailu, Yeneneh T. Belayneh, Henri E. Z. Tonnang, Sunday Ekesi, Sevgan Subramanian.

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
