## [Decision Letter · Decision Letter 0]

7 Dec 2020

PONE-D-20-31046

Bioecology of fall armyworm Spodoptera frugiperda (J. E. Smith), its management and potential migration pattern in Africa

PLOS ONE

Dear Dr. Niassy,

Thank you for submitting your manuscript to PLOS ONE. After careful consideration by three expert reviewers and myself, we recognize the relevance of the data presented, but we also consider that in its current form does not fully meet PLOS ONE’s publication criteria. Therefore, we invite you to submit a revised version of the manuscript that addresses the points raised during the review process.

Note that he majority of concerns with the current version relate to overextending of conclusions and claims not being supported by the type of data presented. While my recommendation is for "minor revision", this is the second time that some of these concerns are expressed and you will need to carefully edit your manuscript to address them as suggested by the reviewers. Because the suggested corrections do not require additional experiments, I trust you will be able to appropriately resolve all of the remaining concerns by following the detailed indications of the reviewers.

We look forward to receiving your revised manuscript.

Kind regards,

Juan Luis Jurat-Fuentes

Academic Editor

PLOS ONE

Journal Requirements:

"We thank FAO, which made the data used in this paper freely available. The authors express their gratitude to farmers, extension agents and all technical staff members involved in the data collection in the different countries. USAID/OFDA contributed to the national capacity that enabled the data to be generated. We also gratefully acknowledge the financial support for the core research agenda of ICIPE by UK Aid from the UK Government; the Swedish International Development Cooperation Agency (SIDA); the Swiss Agency for Development and Cooperation (SDC); and the Government of Kenya."

"The current study was funded by USAID through the FAO.

Dr Sevgan Subramanian is the PI at icipe and Dr Saliou Niassy implemented project activities"

4. We note that Figures 1, 2, 9, 10, 11, and 12 in your submission contain map images which may be copyrighted. All PLOS content is published under the Creative Commons Attribution License (CC BY 4.0), which means that the manuscript, images, and Supporting Information files will be freely available online, and any third party is permitted to access, download, copy, distribute, and use these materials in any way, even commercially, with proper attribution. For these reasons, we cannot publish previously copyrighted maps or satellite images created using proprietary data, such as Google software (Google Maps, Street View, and Earth). For more information, see our copyright guidelines: http://journals.plos.org/plosone/s/licenses-and-copyright.

(1) You may seek permission from the original copyright holder of Figures 1, 2, 9, 10, 11, and 12  to publish the content specifically under the CC BY 4.0 license. 

5. Please ensure that you refer to Figure 8 in your text as, if accepted, production will need this reference to link the reader to the figure.

Reviewers' comments:

Reviewer's Responses to Questions

**Comments to the Author**

1. Is the manuscript technically sound, and do the data support the conclusions?

Reviewer #1: Yes

Reviewer #2: Partly

Reviewer #3: Yes

2. Has the statistical analysis been performed appropriately and rigorously? 

Reviewer #1: Yes

Reviewer #2: I Don't Know

Reviewer #3: Yes

3. Have the authors made all data underlying the findings in their manuscript fully available?

Reviewer #1: No

Reviewer #2: Yes

Reviewer #3: Yes

4. Is the manuscript presented in an intelligible fashion and written in standard English?

Reviewer #1: Yes

Reviewer #2: Yes

Reviewer #3: Yes

5. Review Comments to the Author

Reviewer #1: My comments are on an attached Word document. This manuscript provides valuable information for farmers, extension workers, and scientists in east Africa related to fall armyworm. It will give baseline information for future research.

Reviewer #2: This paper documents a monitoring and data collection system via a mobile app across a wide geographic area, spanning multiple countries, a remarkable achievement for IPM. The effort speaks to a technological and sociological effort that appears to be sustainable and adaptable to other species. I think the authors miss an opportunity to emphasize this in the discussion.

The authors mine a rich dataset from this system to describe the current population dynamics of an invasive species that threatens food security and test for variables that influence these dynamics. There is a pressing need to get this information out to a wide audience, and I think this paper should be published. The dynamics of invasive species may well change, thus the dataset has historical value.

The language about testing the influence of explanatory variables with a negative binomial (NB) model is confusing. My understanding is that a NB model is used to fit a probability density function, or a frequency distribution, and I understand why this would be a good choice when working with count data. However, a NB model does not test for significance of explanatory variables. For example, Tables 1-3, report regression estimates and their significance. From reading the responses to previous reviews, it says the authors used “…generalized linear model that assume a negative binomial distribution error …. This model uses the MASS package which has been included in the ‘Data analysis section’ “This is useful – but it is NOT in the data analysis section!

Briefly explain what IRR refers to in the section on statistical methods, and how to interpret this statistic. Also, clarify that you are developing this metric for the data across the entire time frame of this dataset (at least, that’s what I think you are doing, right?)

What are the sample sizes for the analyses in Tables 1-3. Please give the degrees-of-freedom for the regression estimates in Tables 1-3. I am trying to figure out if I am looking at tests using thousands of datapoints, or if the data were organized into means (per site, month, whatever?) prior to running the regression.

Lines 232-234 – why did you not include beans since it also has a significant regression estimate (albeit at only one *).

Lines 246-247: you claim push-pull reduces infestation relative to seasonal, but Table 2 shows a ns regression estimate and an IRR of almost 1. If I am reading Table 2 correctly, you were NOT able to discern a significant effect from push-pull. The 95% CI for the IRR, however, is very large. Perhaps the conclusion is that you see wide variation in the effect of push-pull, perhaps due to a small sample size?

Line 252-253 – why are you including cassava and peanut, when Table 3 shows a ns regression estimate?

Please report the units for the variables of trap count and do this throughout the manuscript and in the tables and figures. Moths/trap/week? Mean of moths/trap/week across some spatial area? Means of moth/trap/month (which I think is the case for figure 7?) What about Figure 8? Readers need to understand the time component of the unit (week? month? entire study dataset?) for each analysis / figure / table. I found this text in the response to earlier reviews: “Trap counts (adult counts) is the total number of moths in traps. Scouting count refers to the number of infested plants recorded in a sample of 50 plants per field.” This is helpful – but the analyses/tables/figures lead me to believe that some of the time I am looking at means or sums – this needs to be clarified.

The same comment for the variable ‘scouting data’ – what units we are looking at? Are the data being summed, or averaged, over some unit of space or time? It seems to vary at different parts of the text and figures/tables.

Figure 8 is not mentioned in the text. Also, what are the units in the y-axis of Figure 8 referring to? They are in values of thousands, and the axis is labelled as ‘crop stage’. What does that refer to?

The terms ‘seasonal cropping’ and ‘crop rotation’ are being used in a way that is different than other parts of the world. To clarify, include the explanation that you placed in the response to reviewers in the intro (Seasonal cropping is a farming practice in which the same crop is grown in the same area for a number of growing seasons. In the case of maize, it is mainly rain fed, and the land remain fallow between seasons. While crop rotation is the practice of growing a series of different types of crops in the same area across seasons.)

For the UniTraps buckets – did you use the all green version, or the yellow/white/green version, or was this unknown?

Lines 162-163 – give the Latin name for the Desmodium spp. and Napier grass, if possible.

Line 190 – the word data is repeated

Lines 341-342 – the sentence that starts with ‘Sorghum…” is not a sentence.

Line 467 – remove ‘on’

Results section 3.3 – explain the terms ‘Meher’ and ‘Belg’

Reviewer #3: This manuscript is much improved over earlier iterations, and I believe it is suitable for publication from my point of view after considering the comments below. My major concern is still the following, even though the authors have partially addressed the problem:

L204-207: The methodology described does not and cannot “determine FAW migratory patterns at the regional level”, nor can it “indicate the movement of the adult moths across the region.” The trap counts and heat maps are generated from sampling data and only show the change in FAW density over time in each location, and differences in FAW density between locations at a point in time. These data by themselves cannot distinguish whether density changes over time and density differences across space are caused by migration or by population growth/decline from local reproduction and mortality. One or the other or a combination of both could be occurring. Some of the responses to past reviews at first made me think the authors understand this in principle, but sentences like these suggest maybe not. It is known that FAW is a migratory species in North America. Everybody knows this is a migratory species. But no one knows yet how that migratory behavior (which is complicated enough in the Americas) translates to Africa or Asia. Given the rate of spread, FAW is surely exhibiting migratory behavior in its new hemisphere, but what that looks like now in Africa is unknown. Demonstrating patterns and timing to and from different regions on the newly invaded continent of Africa will require additional kinds of data like radar, high-altitude sampling, back-track wind trajectory analyses, and so forth. Heat maps and pheromone trap captures can contribute to figuring out migratory patterns, mainly by generating hypotheses to test, but they simply cannot do so the way the authors seem to think. In my opinion, the authors must use more careful language; for example, they can say their data provide clues to possible migratory patterns, and form the basis of hypotheses about movement patterns that can be tested. Generating meaningful hypotheses is not a small thing; good hypotheses are essential to making progress by not wasting time and resources on more speculative conjectures; the authors should emphasize this because this will be a major impact of their data. But they cannot say the heat maps and trap data “indicate the movement” of the moths or “determine migratory patterns” at any scale. It is possible the heat maps and trap data reflect such movement, but we cannot know this is true without additional, independent kinds of data. It will take a big effort (probably of multiple multidisciplinary teams over several years) of gathering and analyzing other kinds of data to eventually clarify patterns of FAW migratory movement in Africa. Because of these considerations, and because “migration” is a very specific yet complex kind of movement, I also suggest the authors change “potential migration pattern” in the title to “potential patterns of seasonal spread” (echoing verbiage on line 85), which is more defensible based on the data presented in this paper, and will be of equal interest to those concerned about FAW management.

Minor comments:

L35: change “Have implication on” to “impact”

L50: change “crop” to “crops”

L52: change “biotic constrants” to “biotic factors”

L155: delete repeated instance of “and field scouting data”

L190: delete first instance of “date”

L209: I encourage the authors to use a different term than “intersect sites”, because it is not intuitive what this means, even though they define here how they will be using it in this paper. What the authors are describing is a region of year-round reproduction. Alternative, more intuitively descriptive terms for this could be: “permanent distribution” or “year-round distribution” or “area of year-round residency”.

L213: “caters of overdispersion in integer counts data” This phrase is unclear. Suggest changing to “accommodates overdispersion of integer count data”.

L219: delete first instance of “a”

L232: delete hyphen from “Like-lihood”

L302: change “larvae (scouting)” to “larval scouting”

L326: change “built” to “build”

L37, 332, 362: change “larvae” to “larval”

L341-343: change to “Data suggest more attention is needed on other crops attacked by FAW such as sorghum (32), Africa’s second most important cereal in terms of tonnage (31).

L466-468: “the possibility of a sporadic infestation can be speculated on in Africa compared to the Americas where FAW faces winter” The meaning of this whole phrase is unclear. What is meant by “sporadic infestation”? What does FAW “facing winter” in the Americas have to do with anything? The intended meaning and the logic are not clear.

L488: delete “in”

L677: change “Scounted” to “scouted”

6. PLOS authors have the option to publish the peer review history of their article (what does this mean?). If published, this will include your full peer review and any attached files.

Reviewer #1: No

Reviewer #2: No

Reviewer #3: No

---

## [Author Response · Author response to Decision Letter 0]

28 Jan 2021

We have provided a point-by-point response file to all reviewers comments. New changes have been indicated in blue and an MS with track change has been provided.

---

## [Decision Letter · Decision Letter 1]

19 Feb 2021

PONE-D-20-31046R1

Bioecology of fall armyworm Spodoptera frugiperda (J. E. Smith), its management and potential migration pattern in Africa

PLOS ONE

Dear Dr. Niassy,

Thank you for submitting your revised manuscript to PLOS ONE. We appreciate all the work to address previously raised concerns. However, there is concern related to the experimental set up used and whether it can allow extracting conclusions on armyworm migration. This concerns needs to be addressed. Therefore, we invite you to submit a revised version of the manuscript that addresses this point. You can find more specific details in the reviewer's comments. We consider this is a relevant contribution to the field and look forward to a revised version.

We look forward to receiving your revised manuscript.

Kind regards,

Juan Luis Jurat-Fuentes

Academic Editor

PLOS ONE

Reviewers' comments:

Reviewer's Responses to Questions

**Comments to the Author**

1. If the authors have adequately addressed your comments raised in a previous round of review and you feel that this manuscript is now acceptable for publication, you may indicate that here to bypass the “Comments to the Author” section, enter your conflict of interest statement in the “Confidential to Editor” section, and submit your "Accept" recommendation.

Reviewer #1: All comments have been addressed

Reviewer #2: All comments have been addressed

Reviewer #3: (No Response)

2. Is the manuscript technically sound, and do the data support the conclusions?

Reviewer #1: (No Response)

Reviewer #2: (No Response)

Reviewer #3: Partly

3. Has the statistical analysis been performed appropriately and rigorously? 

Reviewer #1: (No Response)

Reviewer #2: (No Response)

Reviewer #3: N/A

4. Have the authors made all data underlying the findings in their manuscript fully available?

Reviewer #1: (No Response)

Reviewer #2: (No Response)

Reviewer #3: Yes

5. Is the manuscript presented in an intelligible fashion and written in standard English?

Reviewer #1: (No Response)

Reviewer #2: (No Response)

Reviewer #3: Yes

6. Review Comments to the Author

Reviewer #1: (No Response)

Reviewer #2: (No Response)

Reviewer #3: The authors have not addressed the main concern I voiced last time. I've pasted it again here, because they simply cannot say (Lines 209-213): "Further, data collected from pheromone traps (universal bucket traps) in the eastern and other African countries were quarterly mapped to determine FAW adults migratory patterns at the regional level. Specifically, we overlaid the quarter FAW density heat maps in 2018 and 2019 to indicate the movement of the adult moths across the region." Pheromone trap captures mapped in this way cannot indicate adult migratory patterns, and the density heat maps cannot indicate the movement of adult moths across the region. I explained previously why these assumptions are not supportable. I feel like we are talking past each other. Assuring me as the reviewer that "more efforts using other approaches are being made" misses the point entirely, because that does not make the assumptions underlying their statements on Lines 209-213 true. The assumptions remain incorrect. It also is not about the authors "making a humble contribution to IPM research on FAW"; I am happy they are making this contribution. The problem is that pheromone trap captures and mapping cannot "determine FAW adult migratory patterns." They simply cannot for the reasons I described previously (pasted here again):

"The methodology described does not and cannot “determine FAW migratory patterns at the regional level”, nor can it “indicate the movement of the adult moths across the region.” The trap counts and heat maps are generated from sampling data and only show the change in FAW density over time in each location, and differences in FAW density between locations at a point in time. These data by themselves cannot distinguish whether density changes over time and density differences across space are caused by migration or by population growth/decline from local reproduction and mortality. One or the other or a combination of both could be occurring. Some of the responses to past reviews at first made me think the authors understand this in principle, but sentences like these suggest maybe not. It is known that FAW is a migratory species in North America. Everybody knows this is a migratory species. But no one knows yet how that migratory behavior (which is complicated enough in the Americas) translates to Africa or Asia. Given the rate of spread, FAW is surely exhibiting migratory behavior in its new hemisphere, but what that looks like now in Africa is unknown. Demonstrating patterns and timing to and from different regions on the newly invaded continent of Africa will require additional kinds of data like radar, high-altitude sampling, back-track wind trajectory analyses, and so forth. Heat maps and pheromone trap captures can contribute to figuring out migratory patterns, mainly by generating hypotheses to test, but they simply cannot do so the way the authors seem to think. In my opinion, the authors must use more careful language; for example, they can say their data provide clues to possible migratory patterns, and form the basis of hypotheses about movement patterns that can be tested. Generating meaningful hypotheses is not a small thing; good hypotheses are essential to making progress by not wasting time and resources on more speculative conjectures; the authors should emphasize this because this will be a major impact of their data. But they cannot say the heat maps and trap data “indicate the movement” of the moths or “determine migratory patterns” at any scale. It is possible the heat maps and trap data reflect such movement, but we cannot know this is true without additional, independent kinds of data. It will take a big effort (probably of multiple multidisciplinary teams over several years) of gathering and analyzing other kinds of data to eventually clarify patterns of FAW migratory movement in Africa. Because of these considerations, and because “migration” is a very specific yet complex kind of movement, I also suggest the authors change “potential migration pattern” in the title to “potential patterns of seasonal spread” (echoing verbiage on line 85), which is more defensible based on the data presented in this paper, and will be of equal interest to those concerned about FAW management."

Why can the authors not change the wording so that they do not claim so much? What they claim in this regard is not defensible in my opinion. Not claiming that they now know the migration patterns in this part of Africa based on pheromone trap captures does not diminish the importance of their data. Instead, asserting that they now do know the migration patterns detracts from the data's importance by making such a mistake in interpretation. Changing the wording to say that the patterns they see in the pheromone trap captures and heat maps "suggest" that the FAW "may be" migrating in such-and-such a pattern in Africa would be fine. It is also fine to have the arrows in Fig. 12 indicating "Potential" migration routes, because this acknowledges that these potential routes are basically hypotheses to be tested, rather than conclusive findings regarding migration routes.

I would add that the sentence (L495-497) "FAW migratory scenario in Africa is different from that in North America where it is established that migration starts from Texas and Florida in the late winter or spring (49)" should be more carefully stated. This is because (1) of course it is different since Texas and Florida are not in Africa, making this a trivial/meaningless statement, and (2) they do not know what the migration pattern in Africa is. Their trapping and density data suggest hypotheses to test (such as in Fig. 12). I do not know what point the authors are trying to make by discussing Texas and Florida as the source for spring migrants to the north in North America, so they will need to clarify that point. Maybe something about differences in seasonality on the two continents? Differences in directionality because of North/South hemispheric differences in seasons? Based on the context of the sentence, when they say "migratory scenario" perhaps they mean to imply this is a "potential" scenario. If so (and I hope so), they need to say this explicitly, because this intent is not conveyed by the current language.

I don't know what else I am supposed to say. This is my best judgment. I hope they can make these simple but important changes in wording. The authors have addressed my other more minor concerns.

7. PLOS authors have the option to publish the peer review history of their article (what does this mean?). If published, this will include your full peer review and any attached files.

Reviewer #1: No

Reviewer #2: No

Reviewer #3: No

---

## [Author Response · Author response to Decision Letter 1]

8 Mar 2021

Dear Editor in Chief,

We have carefully reviewed the comments from the reviewer and provided a point-by-point response document. Key points raised by the reviewer include the fact that the authors claimed to have resolved the FAW migratory pattern in Africa. We want to acknowledge the reviewer’s remarks and comments that adhere to the study's principal objective and conclusion. Therefore, we have revised or removed statements alluding to such claims and modified the title “Bioecology of fall armyworm Spodoptera frugiperda (J. E. Smith), its management and potential patterns of seasonal spread in Africa” as suggested by the reviewer. All statements referring to its migratory pattern are hypothetical with careful language. As the reviewer suggested, more advance tools will be used to test such a hypothesis in subsequent studies. However, the study's main objective was to understand the bioecology of the invasive FAW to implement an effective and sustainable IPM strategy in Africa. The paper recommends ownership of the Community Based Monitoring approach.

We thank the reviewer for his expertise and brilliant analysis and suggestions, which we believe made the revised version more accurate.

Saliou Niassy

Corresponding author

---

## [Editor Report · Decision Letter 2]

10 Mar 2021

Bioecology of fall armyworm Spodoptera frugiperda (J. E. Smith), its management and potential patterns of seasonal spread in Africa

PONE-D-20-31046R2

Dear Dr. Niassy,

We’re pleased to inform you that your manuscript has been judged scientifically suitable for publication and will be formally accepted for publication once it meets all outstanding technical requirements.

NOTE: the statement in lines 489-490 appears unfinished and you may have the chance to edit it during the proofing process.

Kind regards,

Juan Luis Jurat-Fuentes

Academic Editor

PLOS ONE
---

## [Editor Report · Acceptance letter]

20 May 2021

PONE-D-20-31046R2 

Bioecology of fall armyworm *Spodoptera frugiperda* (J. E. Smith), its management and potential patterns of seasonal spread in Africa 

Dear Dr. Niassy:

I'm pleased to inform you that your manuscript has been deemed suitable for publication in PLOS ONE. Congratulations! Your manuscript is now with our production department. 

Kind regards, 

on behalf of

Dr. Juan Luis Jurat-Fuentes 

Academic Editor

PLOS ONE